# A hidden web of policy influence: The pharmaceutical industry's engagement with UK's All-Party Parliamentary Groups

**Emily Rickard**, **Piotr Ozieranski**\*

Department of Social and Policy Sciences, University of Bath, Bath, Somerset, United Kingdom

\* po239@bath.ac.uk

## Abstract

Our objective was to examine conflicts of interest between the UK's health-focused All-Party Parliamentary Groups (APPGs) and the pharmaceutical industry between 2012 and 2018. APPGs are informal cross-party groups revolving around a particular topic run by and for Members of the UK's Houses of Commons and Lords. They facilitate engagement between parliamentarians and external organisations, disseminate knowledge, and generate debate through meetings, publications, and events. We identified APPGs focusing on physical or mental health, wellbeing, health care, or treatment and extracted details of their payments from external donors disclosed on the Register for All-Party Parliamentary Groups. We identified all donors which were pharmaceutical companies and pharmaceutical industry-funded patient organisations. We established that sixteen of 146 (11%) health-related APPGs had conflicts of interest indicated by reporting payments from thirty-five pharmaceutical companies worth £1,211,345.81 (16.6% of the £7,283,414.90 received by all health-related APPGs). Two APPGs (Health and Cancer) received more than half of the total value provided by drug companies. Fifty APPGs also had received payments from patient organisations with conflicts of interest, indicated by reporting 304 payments worth £986,054.94 from 57 (of 84) patient organisations which had received £27,883,556.3 from pharmaceutical companies across the same period. In total, drug companies and drug industry-funded patient organisations provided a combined total of £2,197,400.75 (30.2% of all funding received by health-related APPGs) and 468 (of 1,177–39.7%) payments to 58 (of 146–39.7%) health-related APPGs, with the APPG for Cancer receiving the most funding. In conclusion, we found evidence of conflicts of interests through APPGs receiving substantial income from pharmaceutical companies. Policy influence exerted by the pharmaceutical industry needs to be examined holistically, with an emphasis on relationships between actors potentially playing part in its lobbying campaigns. We also suggest ways of improving transparency of payment reporting by APPGs and pharmaceutical companies.

**Data Availability Statement:** All data relevant to the study are shared in the form of an Excel database available from the University of Bath Research Data Archive. The reference for this dataset is: Rickard, E., Ozieranski, P., in press. Data

set for "A hidden web of policy influence: The pharmaceutical industry's engagement with UK's All-Party Parliamentary Groups (2012-2018)". Bath: University of Bath Research Data Archive. https://doi.org/10.15125/BATH-00943.

**Funding:** Emily Rickard has a +3 PhD Studentship award match-funded (50%) by the Economic and Social Research Council and the University of Bath. This research forms part of the PhD project. PO's work was supported by grants from The Swedish Research Council for Health, Working Life and Welfare (FORTE), no. 2016-00875, and The Swedish Research Council (VR), no. 2020-01822. The funding bodies have played no part in the design or conduct of this study.

**Competing interests:** ER has no conflicts of interests to declare. PO's PhD student was supported by a grant from Sigma Pharmaceuticals, a UK pharmacy wholesaler and distributor (not a pharmaceutical company). The PhD work funded by Sigma Pharmaceuticals is unrelated to the subject of this paper. This does not alter our adherence to PLOS ONE policies on sharing data and materials.

## Introduction

Concerns have long been raised that wide-ranging financial ties to the pharmaceutical industry risk unduly influencing professional judgements [1]. These risks are prevalent in the context of *individual* conflicts of interest in scientific and policy decision-making, including payments to members of expert advisory panels [2,3], scientific article authors [4], and physicians [5,6], as well as *institutional* conflicts of interests through payments to seemingly independent third parties [7] such as patient organisations [8]. These widespread strategies form part of the pharmaceutical industry's 'web of influence' [9,10] seeking to shape the ideologies of the individuals and institutions they fund [11–13] to protect companies' commercial interests, often at the expense of patient health outcomes [14].

One under-explored area of lobbying and institutional conflicts of interest in the UK are All-Party Parliamentary Groups (henceforth APPGs). These informal cross-party groups revolve around a particular topic and are run by and for Members of the UK's Houses of Commons and Lords [15]. They facilitate engagement between parliamentarians and external organisations, providing expertise on complex policy matters, hosting events, and publishing outputs including reports and inquiries. They often have a 'secretariat', an organisation providing administrative support, to facilitate their functioning. APPGs do not receive any funding from Parliament, but they can choose to accept payments to cover costs of events, secretariats, travel, reports, and other activities. APPGs are required to register with the Parliamentary Commissioner for Standards and maintain transparency through declaring payments from external donors. If consultancies or charities act as Secretariats they "must be prepared to disclose information" [16] about their clients or donors, respectively, either on their own websites or on request. However, with the exception of spending regulations on campaigning to support a particular party and prohibitions on paid advocacy specific to individual APPG members (rather than the APPG itself), they are subject to very few regulations regarding their activities or funding [16].

The important role of APPGs is reflected in their increasing involvement in many areas of public health and health policy, including regulations on Fixed Odds Betting Terminals [17] and legislative measures on standardised cigarette packaging [18]. However, concerns have been raised that some corporate interests exploit the unique opportunities for access offered by APPGs, turning them into a backchannel for lobbying [19], or a 'dark space for covert lobbying' [20]. For example, the beer industry [21] and the vaping industry [22], as well as commercial lobbyists more broadly [23], have used APPGs to pursue their policy goals. In particular, lobbyists sometimes act as secretariats for APPGs to gain privileged informal access to legislators [24]. Indeed, some Members of Parliament continue to question whether corporate funding should be allowed at all [25]. These problems are being investigated via an ongoing Parliamentary inquiry [26].

To our knowledge, no bodies analogous to APPGs exist in other contexts, however previous research has documented how the pharmaceutical industry engages with other political landscapes, for example Parliament in Poland [13] and congress in the US [14,27]. The majority of research, however, has prioritised drug companies' 'downstream' lobbying tactics, targeting, in particular, expert advisory bodies or public payer institutions taking decisions on specific drug therapies [28–32]. With few exceptions [13], we know little about how the industry engages with the 'upstream' of the policy process, that is the bodies setting the 'rules of the game' for those at the 'downstream' level, an area frequently lobbied by the tobacco industry [33].

We explore payments received by health-related APPGs from pharmaceutical companies and industry-funded patient organisations between 2012 and 2018. We suggest that, in the context of health related APPGs, payments from the pharmaceutical industry represent

institutional conflicts of interest as they create circumstances where the primary interest (policymaking in the interests of public health) is at risk of being unduly influenced by the secondary interest (the pharmaceutical industry's goal of maximising profits). We also offer a new approach to understanding the potential reach of conflicts of interests by exploring payments to APPGs from organisations with a conflict of interest (namely patient organisations which have received funding from the pharmaceutical industry). Although APPG regulations require patient organisations to disclose funding sources on their website or on request, we know they sometimes underreport payments [34]. Further, this information is not disclosed by APPGs when reporting payments from patient organisations, meaning that APPG members and the public might not be aware of these coinciding conflicts. Broadening the examination of pharmaceutical industry conflicts of interests and how they relate to policy influence is important as, previously, actors involved in industry lobbying strategies have been considered in isolation as these data are disclosed in isolation. We examine patient organisations specifically given the potential they have as important policy vehicles [35,36], particularly as they have been known to put industry interests before patients [8,34,37,38] and receive often substantial industry funds [8]. Overall, we examine how the pharmaceutical industry and the organisations they fund interact with Parliament to form part of a multi-layered web of influence [10].

## Methods

This section explains the process of data collection and analysis for our cross-sectional study focusing on 146 health-related APPGs. The study did not require ethical approval (as it draws on publicly available data at the organisational level), however it is part of a bigger project which has ethical approval from the University of Bath's Social Sciences Research Ethics Committee (approval code: S19-073).

### Data sources

We used two publicly available data sources, both of which have been made accessible to increase transparency and accountability of, separately, APPGs, patient organisations and drug companies. Firstly, the UK Parliament's Register of All-Parliamentary Groups (henceforth the APPG Register) which was first introduced in 2010, with guidelines to increase transparency published in 2015 and updated in 2017. Secondly, drug company disclosure reports of payments to patient organisations which were made mandatory in 2012. We have complete data for drug company payment disclosure reports for 2012–2018, and therefore consider payments registered by APPGs between 2012–2018. Although these data are publicly accessible, consistent with findings from a content analysis of European disclosures of industry payments to healthcare professionals [39] they have limited usability. Therefore, we had to create Excel databases to facilitate analysis—we detail our approach to this below.

To prepare the APPG data, we first created a list of all APPGs with at least one Register entry between 2015–2019 (henceforth the active Register). We included 2019 at this stage in case APPGs were newly registered in 2019 but received payments in 2018. Next, we standardised APPG names (to correct typos or name changes), with a total of 888 unique APPGs identified. We then identified all APPGs with a health focus (see Fig 1 for specific inclusion criteria), ensuring no relevant APPGs were missed by considering definitions of illnesses treatable through pharmaceutical interventions that may have broadened due to the pharmaceuticalisation process [40]. We identified 146 health-related APPGs. At the time of data collection, Parliament only held Register's from 30[th] July 2015 onwards. To access older Registers, we used *the WayBackMachine* web archive to identify any entries between 2012–2015 (henceforth the archived Register)– 91 were registered. ER extracted all available information on payments

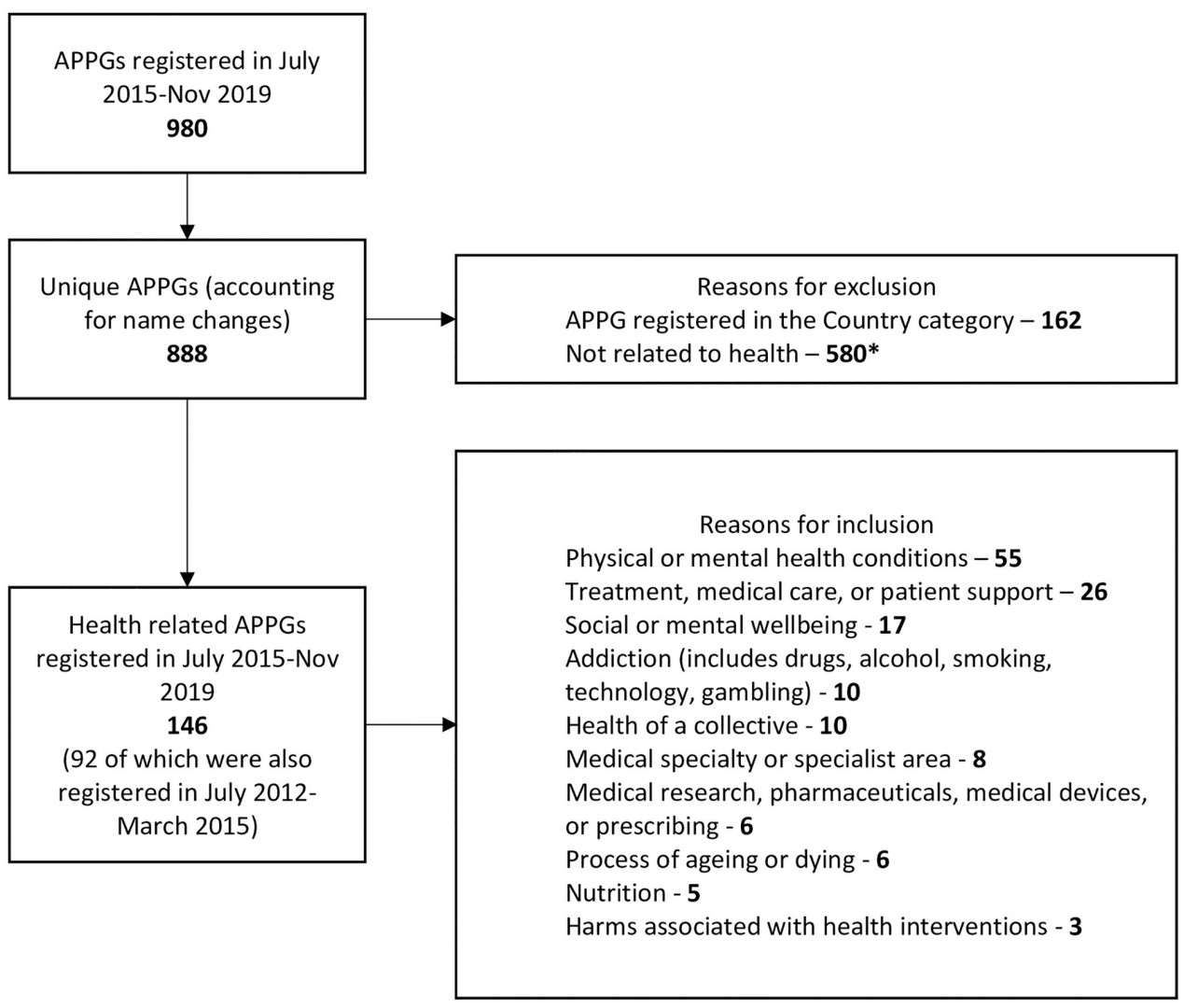

**Fig 1. Identifying relevant APPGs.** * Sport and other recreational activities (45), Faith, culture, or identity (40), Transport—airports, trains, roads (36), Finance or banking (29), Cities, regions, or places (27) Environment, conservation, or sustainability (24), Education—school, university, subjects (22), Energy, chemicals, oils or gases (22), Children or young people (20), Infrastructure, housing or property (19), Defence or security (18), Crime, corruption, or policing (17), Trade or Brexit (15), Equality, diversity, or human rights (13), Technology or media (13), TV, radio, or music (13), Business (12), Law (12), Parliamentary processes, government, or councils (11), Animals and animal welfare (10), International collaborations and relations or foreign affairs (10), Economic or social growth and development (9), People and society (9), Employment (8), Agriculture and farming (7), Art or other creative activities (7), Industries—manufacturing, engineering, manual labour (7), Charities or charitable work (6), Communities and societies (6), Democracy (6), Migration and immigration (6), Science, policy, or analysis (6), Water (6), Family (5), Food and drink —not health focused (5), Governance (5), Hospitality—e.g. catering, tourism (5), Public spaces or services (5), Retail and sales (5), Sexual or domestic violence (5), Alcohol (4), Industries—raw materials (4), Ports, ships, or sailing (4), Poverty (4), Publishing, design, printing, or writing (4), Safety or safeguarding (4), Beauty and fashion (3), Adoption or fostering (2), Events (2), History (2), Taxation (1).

provided by all donors from the active (in Jan-March 2020) and archived (in April-May 2020) Registers into an Excel database.

The database of APPG payments was screened to exclude payments entered more than once (payments are disclosed for 12 months therefore payments may have been extracted multiple times). These were identified through merging columns and identifying duplicates using Excel, followed by manual verification. Excluding duplicates and payments registered before

2012 (in the archived Register) or received after 2018 (in the active Register) resulted in a database of 1,177 payments from all external donors reported between 2012–2018 (see S1 Appendix for an overview of the approach to identifying payments). All payments were checked against the original Register entry. Names of organisations providing payments to APPGs were standardised (through web searches).

For the top five APPG recipients of pharmaceutical industry funding, we supplemented the information provided in the Register with information on their websites to exemplify the importance of APGPs for industry. We chose the top 5 as combined they received over three quarters of the total value provided by industry. We identified all publications available on their websites and coded them by their type (e.g. report), indicated whether industry was involved financially (providing funding for the publication) or non-financially (such as providing a written contribution), and extracted the key topics which were the focus of the publications.

Our second data source, annual disclosure reports detailing drug company payments to patient organisations, are published annually on drug company websites. The data was collected in two waves. ER collected disclosures covering 2012–2016 in June 2017-July 2018 and disclosures covering 2017–2018 –in June-August 2019 –and manually extracted the data to create a database of payments. The database includes 7,023 payments worth £91,443,284.67 to 621 patient organisations, with data extraction detailed elsewhere [8,37]. Briefly, the database contains pharmaceutical company donors, patient organisation recipients, payment dates, descriptions, and values.

### Data management and analysis

All registered APPGs are required to report details of any support received from external sources–this can be a financial benefit or a benefit in kind—if the total value from that source exceeds £1,500 in the calendar year. Financial benefits (henceforth *financial payments*), monetary payments to the APPG, are disclosed with the name the donor and the payment value. Benefits in kind (henceforth *in-kind payments*) involve providing goods or services to APPGs, for example funding events or membership fees, and are disclosed with the name of the donor and a brief description. Prior to July 2015, in-kind payments did not consistently have a payment value disclosed, but since July 2015 in-kind payments have been disclosed with approximate value in bands of £1,500 (e.g. £15,001–16,500). In these cases, bracket averages were calculated as the approximate value (e.g. £15,751). In the results we at times distinguish between financial and in-kind payments as they are subject to different reporting requirements (namely the payment descriptions). As payment values are not always provided for data from the archived Register, we consistently provide the number of payments alongside values.

Separately, payments can be *direct* (donor provides payment to an APPG) or *indirect* (one or more donor funds a third-party to provide a payment to an APPG). As an indirect payment can be funded by multiple donors, we refer to these as 'contributions' throughout the results section to capture that one indirect payment can be funded by multiple organisations. When an indirect payment received multiple contributions, we determined an estimated relative value from each donor by dividing the payment value by the number of donors behind it. For example, if four donors contributed to an indirect payment, and three of them were pharmaceutical companies, the pharmaceutical companies' contribution was calculated as 25% each and 75% of the total.

We used web searches to identify pharmaceutical companies and patient organisations amongst the donors to identify our final sample of payments. Patient organisations were

looked up in the database of pharmaceutical industry payments to patient organisations, and if there was a match the payment details were extracted into a separate Excel spreadsheet.

All payments are expressed in 2018 GBP based on the Consumer Price Index obtained from the Office for National Statistics. Currencies are converted at the annual rate for the patient organisation data (all APPG figures were already in GBP).

We analysed the data descriptively in Microsoft Excel.

## Results

In total, 120 of 146 (82.2%) health-related APPGs reported 1,177 payments from all external donors, with a total value of £7,283,414.90.

During our initial analysis of the APPG data, we developed a typology of relationships (outlined in Fig 2) detailing the major ways in which pharmaceutical companies may influence Parliament using APPGs. Relationship #1 relates to conflicts of interest through payments directly provided by a pharmaceutical company to an APPG, as disclosed in the APPG Register. In addition, Relationship #2 involves conflicts of interests through indirect payments, that is organisations being funded by pharmaceutical companies with the specific purpose of providing a service for an APPG, as disclosed in the APPG Register. Finally, Relationship #3 covers payments from patient organisations, as disclosed in the APPG Register, which have coincidentally received payments from the pharmaceutical industry (identified in drug company disclosures of payments to patient organisations).

Overall, the pharmaceutical industry and industry-funded patient organisations (relationships #1, #2, and #3) provided a total of £2,197,400.8 (30.2% of the £7,283,414.9 received by health related APPGs) across 468 payments (39.8% of the 1,177 payments received by health related APPGs) to 58 (39.7%) of 146 health-related APPGs (see Table 1). The APPG for Cancer followed by Health were targeted with the highest value and number of payments from pharmaceutical companies and industry-funded patient organisations. We will expand upon the three relationships outlined in Fig 2, unpacking the contents of Table 1 throughout the subsequent subsections.

### Relationship #1: Conflicts of interest through direct pharmaceutical industry payments

Thirty (10.8%) of the 277 donors providing direct payments were pharmaceutical companies. Of a total 1,066 direct payments from all donors with a reported value of £6,268,955.0, industry provided 129 (12.1%) worth £858,647.95 (13.7%). Ten (33.3%) of the thirty drug companies provided £534,332.75 (62.2% of industry's total contribution), indicating high levels of donor concentration (S1 Table provides a list of companies and their payments).

Industry's payments targeted ten APPGs from three categories (Table 2). APPGs categorised as 'physical or mental health conditions' received both the highest number and value of industry payments. Concentration was noted in recipients as well as donors, with industry payments comprising at least 40% of the direct income received by half of the APPGs reporting industry payments. Further, two APPGs (Health and Cancer) received over half of industry's total direct payments.

Seventy-one industry payments worth £513,772.1 were in-kind and therefore disclosed with a description. The most frequent purpose for industry payments was events (33 payments worth £292,175.6) followed closely by membership fees (32 payments worth £292,175.6). The remaining six payments covered four secretariat costs (totalling £32,146.2), one report costs (£8,250.5) and one had two purposes–inquiry staff and event costs (£32.250.5).

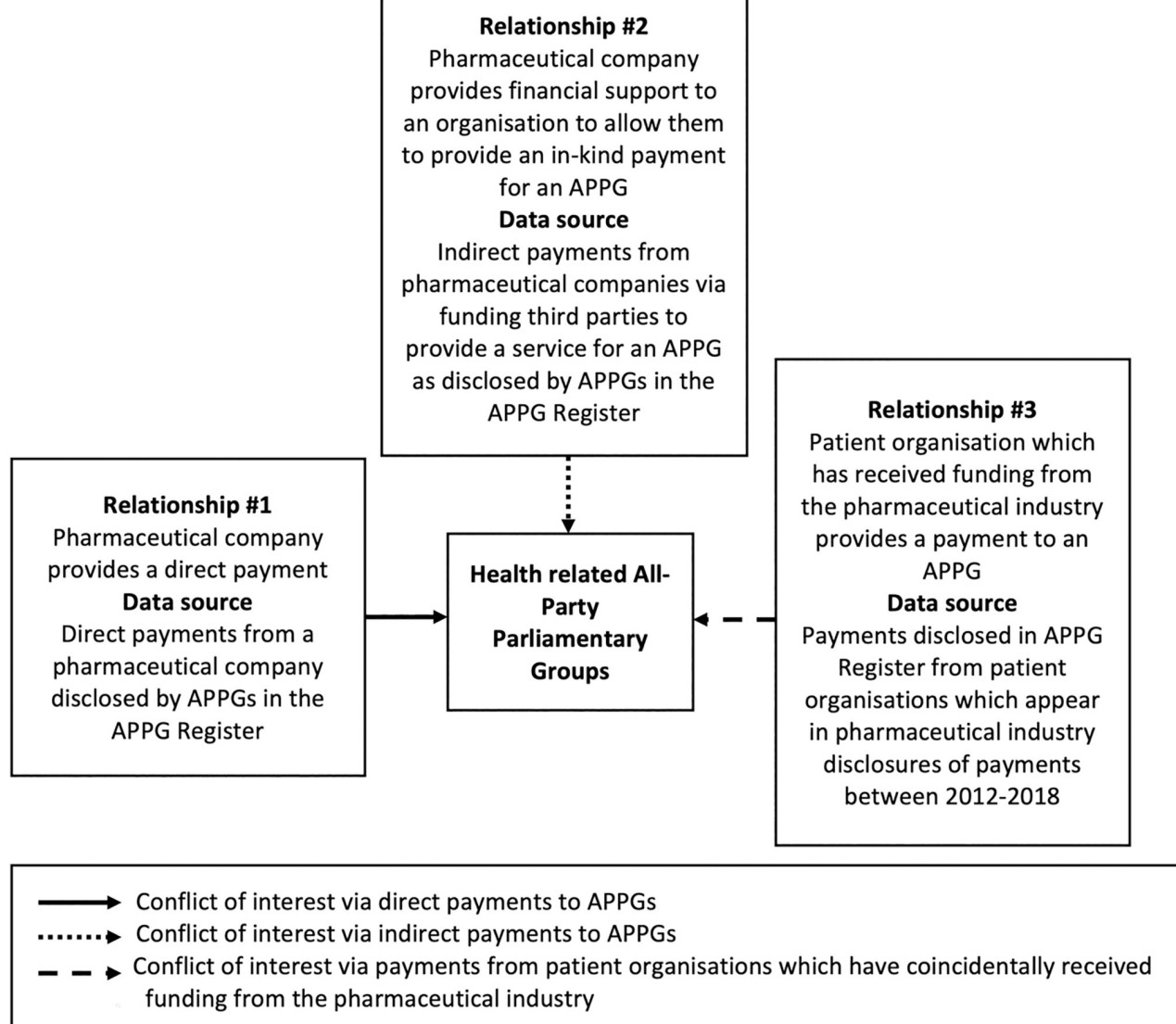

**Fig 2. Different types of ties between pharmaceutical companies and APPGs.**

### Relationship #2: Conflicts of interest through indirect pharmaceutical industry payments

Sixteen (19.5%) of the 82 donors making indirect payments were pharmaceutical companies. Of a total 247 contributions to 111 indirect payments worth £1,014,459.9 from all donors, the pharmaceutical industry made 69 (27.9%) contributions towards 39 (35.1%) indirect payments worth £352,697.9 (34.8%). As with direct payments, indirect payments were concentrated by donors, with the top three drug company donors providing over half of the industry's indirect payments (see S2 Table for a list of companies). Different companies prioritised direct and indirect payments. Overall, industry provided more direct payments than indirect.

Industry's indirect payments targeted nine health-related APPGs from the same three categories as their direct payments (Table 3). The direct and indirect payments both targeted

**Table 1. Overview of all payments received by health related APPGs between 2012–2018.**

| APPG name | Payments from all external sources | | | Payments from pharmaceutical companies and pharmaceutical industry-funded patient organisations | | |
|---|---|---|---|---|---|---|
| | Value of payments—£ | Number of payments—n§ | Number of payments with value—n (%) | Value of payments—£ (%)¶ | Number of payments—n (%)¶ | Number of payments with value—n (%)¶ |
| Cancer‡ | 442,318.21 | 54 | 50 | 440,573.21 (99.61) | 53 (98.15) | 49 (98) |
| Health* | 1,017,516.98 | 108 | 104 | 414,921.47 (40.78) | 47 (43.52) | 47 (45.19) |
| Thrombosis‡ | 224,094.40 | 9 | 5 | 146,545.81 (65.39) | 8 (8.89) | 4 (80) |
| Sickle Cell and Thalassaemia* | 122,527.46 | 16 | 8 | 122,527.46 (100) | 16 (100) | 8 (100) |
| Sepsis† | 154,979.29 | 11 | 8 | 71,441.14 (46.10) | 3 (27.27) | 3 (37.50) |
| HIV and AIDS* | 329,525.96 | 42 | 41 | 66,083.63 (20.05) | 7 (16.67) | 7 (17.07) |
| Rare, Genetic and Undiagnosed Conditions† | 65,595.93 | 4 | 4 | 65,595.93 (100) | 4 (100) | 4 (100) |
| Dementia† | 64,296.82 | 17 | 13 | 64,296.82 (100) | 17 (100) | 13 (100) |
| Liver Health‡ | 59,906.00 | 10 | 6 | 59,906.00 (100) | 10 (100)* | 6 (100) |
| Obesity* | 94,763.94 | 7 | 3 | 45,531.23 (48.05) | 1 (14.29) | 1 (33.33) |
| Autism† | 45,065.61 | 8 | 4 | 45,065.61 (100) | 8 (100) | 4 (100) |
| Atrial Fibrillation‡ | 51,935.62 | 8 | 4 | 46,297.69 (89.14) | 8 (100) | 4 (100) |
| Alcohol Harm‡ | 51,264.61 | 8 | 4 | 40,014.11 (78.05) | 8 (100) | 3 (75) |
| Sexual and Reproductive Health‡ | 115,768.08 | 16 | 11 | 37,961.99 (32.79) | 8 (10) | 4 (36.36) |
| Women's Health* | 71,223.12 | 3 | 3 | 35,611.56 (50) | 8 (100) | 3 (100) |
| Breast Cancer† | 34,058.30 | 10 | 6 | 34,058.3 (100) | 10 (100) | 6 (100) |
| Pancreatic Cancer† | 33,935.61 | 8 | 4 | 33,935.61 (100) | 8 (100) | 4 (100) |
| Eye Health and Visual Impairment‡ | 32,250.50 | 9 | 1 | 32,250.50 (100) | 5 (55.56) | 1 (100) |
| Brain Tumours† | 46,494.61 | 10 | 4 | 30,997.96 (66.67) | 10 (100) | 4 (100) |
| Multiple Sclerosis† | 29,344.54 | 12 | 4 | 29,344.54 (100) | 12 (100) | 4 (100) |
| Baby Loss† | 29,114.20 | 4 | 4 | 29,114.2 (100) | 4 (100) | 4 (100) |
| Stem Cell Transplantation† | 26,552.76 | 10 | 6 | 26,552.76 (100) | 10 (100) | 6 (100) |
| Skin* | 222,779.70 | 27 | 23 | 23,437.37 (10.52) | 13 (48.15) | 13 (56.52) |
| Ageing and Older People† | 21,698.59 | 8 | 4 | 21,698.59 (100) | 8 (100) | 4 (100) |
| Diabetes‡ | 20,575.70 | 11 | 7 | 20,575.70 (100) | 11 (100) | 7 (100) |
| Muscular Dystrophy† | 19,250.08 | 9 | 5 | 19,250.08 (100) | 9 (100) | 5 (100) |
| Tuberculosis* | 58,650.74 | 19 | 9 | 19,114.06 (32.59) | 5 (26.32) | 4 (44.44) |
| Haemophilia and Contaminated Blood† | 17,168.74 | 10 | 6 | 17,168.74 (100) | 10 (100) | 6 (100) |
| Osteoporosis† | 16,188.16 | 10 | 3 | 16,188.16 (100) | 10 (100) | 3 (100) |
| Young Disabled People† | 15,499.58 | 8 | 4 | 15,499.58 (100) | 4 (50) | 4 (100) |
| Motor Neurone Disease† | 10,693.15 | 8 | 4 | 10,693.15 (100) | 8 (100) | 4 (100) |
| Heart and Circulatory Diseases† | 10,063.56 | 7 | 3 | 10,063.56 (100) | 4 (57.14) | 3 (100) |
| Ovarian Cancer† | 9,300.58 | 8 | 4 | 9,300.58 (100) | 8 (100) | 4 (100) |
| Children, Teenagers, and Young Adults with Cancer† | 9,106.27 | 4 | 4 | 9,106.27 (100) | 4 (100) | 4 (100) |
| Blood Cancer† | 8,489.30 | 3 | 3 | 8,489.3 (100) | 3 (100) | 3 (100) |
| Malaria and Neglected Tropical Diseases* | 270,418.00 | 35 | 32 | 8,250.50 (3.05) | 1 (2.86) | 1 (3.13) |
| 22 other APPGs† | 248,423.16 | 158 | 35 | 69,937.58 (28.15) | 110 (69.62) | 24 (68.57) |

*(Continued)*

**Table 1.** (Continued)

| APPG name | Payments from all external sources | | | Payments from pharmaceutical companies and pharmaceutical industry-funded patient organisations | | |
|---|---|---|---|---|---|---|
| | Value of payments—£ | Number of payments—n§ | Number of payments with value—n (%) | Value of payments —£ (%)¶ | Number of payments—n (%)¶ | Number of payments with value—n (%)¶ |
| Total | 4,100,837.87 | 709 | 443 | 2,197,400.75 (53.58) | 468 (66.01) | 278 (62.75) |

* APPGs receiving payments from pharmaceutical companies only (n = 8).

† APPGs receiving payments from pharmaceutical industry-funded patient organisations (n = 42).

‡ APPG receiving payments from both the pharmaceutical industry and pharmaceutical industry-supported patient organisations (n = 8).

§ Four payments were jointly from a pharmaceutical company and a pharmaceutical industry funded patient organisation–these are counted as four payments rather than eight.

¶ Percentages are the number/value of payments from pharmaceutical companies and pharmaceutical industry-funded patient organisations as a proportion of the total number/value of payments received by each APPG.

APPGs categorised as 'physical or mental health conditions' with the highest number and value of payments. The pharmaceutical industry were big donors within the three categories, providing 47.0% (39 of 83) of all indirect payments received. Similar to the targeted funding identified in the direct payments, over half of industry's indirect payments went to two APPGs (although these were different—Sickle Cell and Thalassaemia and Thrombosis).

**Table 2. Direct payments from pharmaceutical companies received by health related APPGs.**

| APPG category (in bold) and name* | Direct payments from all donors—£ | Direct payments from all donors—n | Direct payments from pharmaceutical industry—£ (%†) | Direct payments from pharmaceutical industry—n (%‡) |
|---|---|---|---|---|
| **Health of a collective** | **1,554,102.11** | **216** | **414,921.47 (26.7)** | **47 (21.76)** |
| Health | 1,017,516.98 | 108 | 414,921.47 (40.78) | 47 (43.52) |
| **Physical or mental health conditions** | **1,806,550.62** | **379** | **358,195.03 (19.83)** | **65 (17.15)** |
| Cancer | 442,318.21 | 54 | 252,557.67 (57.1) | 45 (83.33) |
| HIV and AIDS | 329,525.96 | 42 | 66,083.63 (20.05) | 7 (16.67) |
| Tuberculosis | 58,650.74 | 17 | 19,114.06 (32.59) | 4 (23.53) |
| Malaria and Neglected Tropical Diseases | 270,418.00 | 35 | 8,250.50 (3.05) | 1 (2.86) |
| Diabetes | 20,575.70 | 7 | 7,501.00 (36.46) | 2 (28.57) |
| Sickle Cell and Thalassaemia | 4,688.17 | 6 | 4,688.17 (100) | 6 (100) |
| **Medical specialty or specialist area** | **1,073,272.30** | **130** | **85,531.46 (7.97)** | **17 (17.15)** |
| Eye Health and Visual Impairment | 32,250.5 | 9 | 32,250.50 (100) | 1 (11.11) |
| Liver Health | 59,906.00 | 6 | 29,843.59 (49.82) | 3 (50) |
| Skin | 222,779.70 | 27 | 23,437.37 (10.52) | 13 (48.15) |
| Total received by the APPGs | 2,458,629.96 | 311 | 858,647.95 (34.92) | 129† (41.48) |

* Bold text indicated the category of the APPG to allow us to present the total number and value of payments received by each category as well as each APPG receiving industry payments.

† Percentages are of the total value and number of direct payments received from industry by each APPG.

‡ Four payments did not have a value disclosed.

**Table 3. Total number and value of indirect payments from pharmaceutical companies as a proportion of all indirect payments (2012–2018).**

| APPG category (in bold) and name** | Total indirect payments—n | Total indirect payments with value—n | Total value of indirect payments†—£ | Total indirect payments from pharmaceutical companies—n (%) | Total indirect payments from pharmaceutical companies with value—n (%) | Total value of indirect payments from pharmaceutical companies —£ (%) |
|---|---|---|---|---|---|---|
| **Health of a collective** | **8** | **4** | **87,918.65** | **8 (100)** | **4 (100)** | **52,307.09 (59.49)** |
| Alcohol Harm | 5 | 1 | 16,695.53 | 5 (100) | 1 (100) | 16,695.53 (100) |
| Women's Health | 3 | 3 | 71,223.12 | 3 (100) | 3 (100) | 35,611.56 (50) |
| **Medical speciality or specialist area** | **8** | **1** | **18,285.53** | **5 (62.5)** | **1 (100)** | **9,142.77 (50)** |
| Sexual and Reproductive Health | 1 | 1 | 18,285.53 | 1 (100) | 1 (100) | 9,142.77 (50) |
| Liver Health | 4 | 0 | - | 4 (100) | 0 | - |
| **Physical or mental health conditions** | **67** | **32** | **709,189.71** | **26 (38.81)** | **13 (40.63)** | **291,248.00 (41.07)** |
| Sickle Cell and Thalassaemia | 10 | 6 | 117,839.30 | 10 (100) | 6 (100) | 117,839.30 (100) |
| Thrombosis | 8 | 4 | 162,878.87 | 7 (87.5) | 3 (75) | 85,330.28 (52.39) |
| Tuberculosis | 2 | 0 | - | 1 (50) | 0 | - |
| Obesity | 3 | 1 | 68,296.84 | 1 (33.33) | 1 (100) | 45,531.23 (66.67) |
| Atrial Fibrillation | 7 | 3 | 48,185.12 | 7 (100) | 3 (100) | 42,547.19 (88.3) |
| Total received by the APPGs | 43 | 19 | 503,404.31 | 39 (90.7) | 18 (94.74) | 352,697.86 (70.06) |

* Bold text indicated the category of the APPG to allow us to present the total number and value of payments received by each category as well as each APPG receiving industry payments.

* Values were not provided in the archived Register, therefore these values are for payments registered in July 2015 onwards.

**Table 4. Top 10 pharmaceutical company donors by value of payments.**

| Pharmaceutical company | Value of all payments—£ (%)* | Payments—n (%)* | Payments with value provided—n (%)* |
|---|---|---|---|
| Novartis | 153,046.31 (12.63) | 24 (12.12) | 16 (10.32) |
| Bayer | 94,346.5 (7.79) | 15 (7.58) | 7 (4.52) |
| Pfizer | 76,643.17 (6.33) | 11 (5.56) | 10 (6.45) |
| Bristol-Myers Squibb | 75,734.48 (6.25) | 10 (5.05) | 10 (6.45) |
| Gilead | 68,475.48 (5.65) | 7 (3.54) | 7 (4.52) |
| MSD | 58,240.89 (4.81) | 6 (3.03) | 6 (3.87) |
| Janssen | 54,039 (4.46) | 5 (2.53) | 5 (3.23) |
| Pfizer-BMS Alliance | 52,953.24 (4.37) | 12 (6.06) | 5 (3.23) |
| Sanofi Pasteur MSD | 49,270.7 (4.07) | 7 (3.54) | 7 (4.52) |
| Novo Nordisk | 48,618.49 (4.01) | 5 (2.53) | 5 (3.23) |
| Remaining 25 companies | 480,055.06 (39.63) | 96 (48.48) | 77 (49.68) |
| Total | 1,211,345.23 | 198 | 155 |

* Percentages are the number / value of payments from each company expressed as a proportion of the number of payments from all pharmaceutical companies.

All of the indirect payments were in-kind and mainly covered funding third parties to provide secretariat or administrative services (37 payments with a disclosed value of £328,929.8), with the remaining two payments covering costs of a report (£9,142.8) and an event (£14,625.3). The pharmaceutical industry was a prominent funder at the infrastructural level, providing funding for 37 of the 65 (58.7%) indirectly funded secretariats.

## Relationships #1 and #2: Direct and indirect pharmaceutical company payments combined

The combined value of the pharmaceutical industry's direct and indirect payments was £1,211,345.81 across 168 payments and 198 contributions (see S3 and S4 Tables for full breakdown of these payments at the recipient and donor levels) from 35 drug companies to 16 APPGs. APPGs categorised as 'physical or mental health conditions' received the highest number, 91 (20.4% of 446 payments received by the category), and value, £649,443.02 (25.8% of the total £2,515,740.3 received by the category), of payments. Overall, half of the industry's contributions was directed towards two APPGs–Health and Cancer. Similar patterns of concentration were reflected within pharmaceutical companies, with ten (of 35, 28.6%) companies providing £731,368.26 of the total £1,211,345.81 (60.4%)–see Table 4.

Overall, industry's payments targeted three key purposes: secretariat or administrative support, events, and membership fees, with the highest value of payments going towards secretariat and administrative support - £361,076.06. Pharmaceutical companies dominated payments for events, providing 67.4% (£163,574.6 of £242,881.84) of the funding for this purpose across all APPGs, as well as membership fees (53.4%, £292,175.57 of £547,392.48). Further details of the distribution of payment purposes are provided in S5 Table.

## Relationship #3: Payments from patient organisations with conflicts of interest

In addition to the pharmaceutical industry's payments, the APPG Register reported 50 APPGs receiving 304 payments (all in-kind) worth £986,055.0 from 57 patient organisations which feature as payment recipients in pharmaceutical industry payment disclosures. The industry-funded patient organisations which gave payments to APPGs received £27,883,556.30 across 1,965 payments from 65 pharmaceutical companies between 2012–2018. Table 5 provides a list of the top industry-funded patient organisations ordered by the value of payments they made to APPGs, alongside the number and value of the payments they received from industry.

Similar to drug company payment patterns, the majority of industry-funded patient organisations' payments were to APPGs categorised as 'physical or mental health conditions'– 199, or 65.5% of their 304 payments, worth £785,568.72 (79.7% of the £986,054.9 total value (see S6 Table). The majority of the value of payments (£911,452.7 of £986,054.9, 92.4%) covered secretariat or administrative services and the remainder were for five other purposes (see S7 Table). While Table 5 shows the top industry-funded patient organisations by the value of the payments they provided, in terms of the number of payments the APPGs for Dementia, Mental Health, Multiple Sclerosis and Parkinson's received the most payments–receiving 56 (18.4%) of 304 payments.

Combining the payments made by pharmaceutical companies *and* pharmaceutical industry-funded organisations, their payments overlapped across three categories of APPGs (indicated with an asterisk in Table 6), with the 'physical or mental health conditions' category receiving the highest proportion of their payments from these industry sources (290 of 446, 65.0%).

**Table 5. Top 10 pharmaceutical-industry funded patient organisations by value of payments provided to APPGs.**

| Top 10 patient organisations by value of payments to APPGs | Payments to APPGs —£ | Payments from pharmaceutical companies—£‡ | Payments from pharmaceutical companies—n† |
|---|---|---|---|
| Macmillan Cancer Support | 188,015.54 | 212,629.82 | 59 |
| UK Sepsis Trust | 71,441.14 | 5,115.83 | 2 |
| Genetic Alliance UK | 65,595.93 | 740,362.30 | 86 |
| Alzheimer's Society | 64,296.82 | 411,916.81 | 27 |
| Anticoagulation UK | 61,215.53 | 572,541.22 | 116 |
| National Autistic Society | 45,065.61 | 2,433.17 | 2 |
| Muscular Dystrophy UK | 34,749.66 | 85,774.00 | 11 |
| Breast Cancer Now* | 34,058.30 | 3,841,858.40 | 16 |
| Pancreatic Cancer UK | 33,935.61 | 307,995.82 | 22 |
| Hepatitis C Trust | 30,062.41 | 1,883,400.76 | 115 |
| Remaining 45 patient organisations | 357,618.40 | 19,819,528.2 | 1,509 |

* Created after the merger of Breast Cancer Campaign and Breakthrough Breast Cancer in 2015.

† To give context to these values, the median number of payments received by all patient organisations from pharmaceutical companies is 3.

‡ To give context to these values, the median value of payments received by all patient organisations from pharmaceutical companies is £19,684.56.

## APPG outputs

We also considered the ongoing outputs between 2012–2018 of the top five health-related APPGs receiving payments from industry. Notably, not all APPGs maintain a record of all outputs (for example the APPG for HIV and AIDS state "Here's just a small sample of our inquiries over the years"), therefore we can only review the publications currently available on their websites. The five APPGs had 31 outputs available (19 reports, 6 inquiries, 3 consultation responses and 3 essay collections). Inquiries involve input from external organisations and

**Table 6. Direct and indirect payments from pharmaceutical companies and patient organisations funded by pharmaceutical companies.**

| Category of APPG | Payments from all external sources | | | Payments from pharmaceutical companies and industry-funded patient organisations | | |
|---|---|---|---|---|---|---|
| | Payments— n | Payments with value—n | Value of payments—£ | Payments—n (%)‡ | Payments with value—n (%)‡ | Value of payments —£ (%)‡ |
| Physical or mental health conditions* | 446 | 284 | 2,515,740.33 | 290 (65.02) | 176 (61.97) | 1,435,011.74 (57.04) |
| Health of a collective* | 224 | 204 | 1,642,020.76 | 62 (27.68) | 54 (26.48) | 492,932.68 (30.02) |
| Medical specialty or specialist area* | 138 | 80 | 1,091,557.83 | 46§ (57.50) | 24 (30) | 153,555.86 (14.07) |
| Social or mental wellbeing† | 118 | 55 | 656,443.19 | 23 (19.49) | 9 (16.36) | 48,364.28 (7.37) |
| Treatment, medical care, or patient support | 88 | 43 | 559,425.37 | 24 (27.27) | 8 (18.6) | 35,675.40 (6.38) |
| Process of ageing or dying† | 35 | 16 | 85,505.03 | 15 (42.86) | 7 (43.75) | 31,860.79 (37.26) |
| Medical research, prescribing pharmaceuticals, or the pharmaceutical industry specifically† | 53 | 11 | 303,554.87 | 8 (15.09) | 0 | - |
| Total | 1102 | 693 | 6,854,247.38 | 468 (67.53) | 283 (40.84) | 2,197,400.75 (32.06) |

* Category received payments from pharmaceutical companies *and* patient organisations which had received pharmaceutical industry funding.

† Category received payments from patient organisations which had received pharmaceutical industry funding only.

‡ Percentages are the proportion of the number/value of payments from pharmaceutical companies and industry-funded patient organisations expressed as a proportion of the total number/value of payments from all donors in each category.

§ Four payments were funded by a pharmaceutical company and a patient organisation funded by the pharmaceutical industry–these payments are counted once, i.e. counted as four payments rather than eight.

were published by the Cancer APPG (n = 2) and the HIV and AIDS APPG (n = 4). One Report from the Cancer APPG also included details of external contributions. All seven publications named drug companies with a total of 28 contributions from 13 different companies. Nineteen of the 28 contributors had provided payments to the APPG publishing the report. This suggests that there is a link between providing payments to APPGs and being involved in their activities. It is also important that additional pharmaceutical companies were involved in APPG activities despite not providing any payments, suggesting that, in some instances, the involvement of the pharmaceutical industry extends beyond what is disclosed in the APPG Register. Examples of industry contributions include the APPG for Cancer's enquiry where AstraZeneca raised the issue of access to cancer drugs [41] and the APPG for HIV and AIDS' enquiry within which Gilead, ViiV Healthcare and Janssen argued for tiered pricing, something that is said to promote profits above access to drugs [42,43]. As well as inquiries accepting industry input, APPGs publish reports and essay collections covering issues pertinent to the pharmaceutical industry, for example the Health APPG's essay collection raises the issue of spending caps on drugs [44] and the Cancer APPG provides a response to a consultation on the Cancer Drugs Fund [45]. Incidentally these two APPGs received the most pharmaceutical industry payments. Also, a Health APPG essay collection was funded by four drug companies and a Report from the Sickle Cell and Thalassaemia was funded by Novartis, neither of which were specifically disclosed in the Register.

## Discussion

### Statement of principal findings

We evaluated the web of pharmaceutical industry influence within health-related APPGs, concluding that 35 drug companies (Relationship #1 and #2) were behind 168 payments, worth £1.2m, to 16 APPGs. These payments were higher value than from all donors and were concentrated at the donor and recipient levels, with the APPGs for Cancer and Health targeted with substantial industry funds. Additionally, we provided preliminary evidence, through exploring outputs published by APPGs, that the conflicts of interests may lead to undue influence as companies are able to contribute to inquiries and have their interests reflected in reports, which they occasionally fund. We also explored the potential for an alternative avenue of influence on Parliament via patient organisations with conflicts of interest. The value of industry-funded patient organisation payments (Relationship #3) was £986,054.94 across 304 payments to 50 APPGs, bringing the combined total of Relationships #1, #2 and #3 to £2,197,400.75 across 468 payments. These substantial funds open up risks of "institutional corruption" which manifests when systemic and strategic influence undermines an institution's effectiveness through diverting it from its purpose [46], namely that commercial interests are served above public health [47]. Overall, the industry's web of influence and financial ties with organisations, particularly in the context of public health policy, requires much greater oversight.

### Strengths and limitations

Our study is the first to explore the financial ties between APPGs and the pharmaceutical industry as well as patient organisations funded by the pharmaceutical industry. In so doing, it applies a novel approach by looking at the engagement of industry-funded patient organisations with policy. It also has some limitations. Firstly, reporting requirements changed in July 2015 and, although we ensured no payment's 'date registered' overlapped, we cannot guarantee there were no discrepancies in reporting. Secondly, due to reporting requirements, some payments registered in 2012 may have been received in late 2011. Thirdly, relative values of

indirect payments were calculated where there was more than one organisation involved—this assumes all organisations provided an equal amount which may over- or under-estimate the contribution. Fourthly, in-kind payments are reported in brackets of £1,500, therefore the mean value could be an over- or under-estimation, but the amount will be negligible. Fifthly, there may be other types of coinciding interests beyond patient organisations, for example industry payments to healthcare organisations or universities which make payments to APPGs, which our study may not have captured. Finally, the payments to patient organisations, although made by the majority of the UK pharmaceutical industry [48], might exclude companies not participating in disclosure initiatives or underreporting their payments [34,37].

## Comparison with other studies

Although no research has explored the pharmaceutical industry's engagement with UK Parliament, we can draw comparisons with studies examining its payments to other organisations. For example, the industry provided £57.3m to UK patient organisations (2012–2016) [8], £47.1m to healthcare organisations (2015) [49] and £5m to Clinical Commissioning Groups (2015 and 2016), that is organisations which commission health services funded through general taxation in England. The lower value of payments to Clinical Commissioning Groups than to, for example, secondary care providers (£20.7m [49]), such as National Health Service Trusts (providers of secondary and tertiary care within England's public health system), suggests that, from the industry's perspective, smaller funding does not necessarily reflect the importance of the recipient. Indeed, we know that, in relation to healthcare professionals, small payments can influence prescribing [5,50,51]. The comparatively low value of payments to APPGs might therefore reflect the relatively low cost of networking and opinion-shaping opportunities they offer, which have been identified as the industry's key influence strategy [14,52]. More broadly, the "cost-effectiveness" of APPGs as an influence channel is demonstrated by the sharp contrast with the industry's vast donations to political party committees in the US [27].

Previous research has found patient organisations can feel pressured to align with industry agendas in ways that prioritise commercial over patient interests [53–55], which is concerning in the context of APPGs as the majority of industry-funded patient organisations' payments were for secretariat purposes. Additionally, most of the pharmaceutical industry's indirect payments were to fund third-parties to act as secretariat. These payments represent "infrastructural" conflicts of interests as secretariats help with the general running of APPGs, therefore holding a prime position and offering further scope to influence an APPG's policy agenda.

Payments to APPGs also reflect policy-related payments recorded elsewhere, for example payments to UK patient organisations for policy engagement [8] and frequent payments to Swedish patient organisations covering 'politicians week' [38], a major annual forum for Swedish politics, suggesting that patient organisations may act as a conduit between industry and policy. Additionally, industry has consistently created platforms for networking through payments for events provided to healthcare organisations [49], healthcare professionals [56], Clinical Commissioning Groups [57], and patient organisations [8,38], a pattern which was reflected in payments to APPGs. Despite their prevalence, industry-sponsored events have received criticism as they may offer an opportunity to influence the event's agenda [58,59] and form part of the industry's broader marketing strategy of building relationships with useful actors [11].

Further similarities with previous studies include the concentration of payments within a narrow group of donors [8,49], recipients [8,49], and conditions [8,38]. The targeted nature of funding reflects companies' attempts to secure commercial benefits [60,61]. The APPG for

Cancer was prioritised, reflecting prioritisation of payments to patient organisations in the UK [8], Sweden [38], US [60], Australia [62], and Canada [63], suggesting industry's web of influence is most dominant in this profitable disease area [64]. We can also draw important comparisons with the top 10 drug company donors to APPGs to those making payments to healthcare organisations [49], healthcare professionals [65], and patient organisations [8] in the UK. For example, Novartis and Pfizer target multiple channels of influence in their funding as they were among the top 10 to all organisation types. Companies targeting various healthcare contexts suggests they use multiple channels of access through funding the upstream and downstream parts of the policy process. However, Gilead was only among the top 10 donors to APPGs, suggesting they adopt different strategies of funding, prioritising the upstream part of the policy process. This observation reflects recent policy controversies related to Gilead building a coalition of support for the funding of high-price hepatitis C drugs [66].

## Policy implications and conclusions

As with previous studies examining Clinical Commissioning Groups, which take strategic decisions on the funding of health services [49,57], ours also shows pharmaceutical industry funding at the upstream stages of the policy process. These recipients are not always of immediate interest to health policy researchers, despite their importance. For example, the APPG for Health offers a unique avenue of influence for industry in the broader health policy landscape, facilitating access to agenda-setting on a wider strategic level. Although complex, further research needs to consider the impact of industry's payments, particularly the targeted funding, on legislative activities as we know from previous research that industry payments are influential [67–69]. Indeed, whilst much of the conflict of interest literature in relation to the pharmaceutical industry has focused on individual conflicts [1,70–72], we show the importance of recognising institutional conflicts. Before considering individual legislators, we need to consider the setting in which they work, and this setting may have been shaped by the pharmaceutical industry.

To help manage these institutional conflicts of interests, transparency must be improved at the level of donors (pharmaceutical industry) and recipients (APPGs). Troublingly, pharmaceutical companies are not required to disclose these payments at all and are therefore missing from Disclosure UK [49], a transparency initiative; a handful of others have been mistakenly reported in disclosure reports covering payments to patient organisations [37] (unpublished background calculations). Therefore, APPGs and other Parliamentary organisations should be added explicitly to the disclosure guidelines included in the ABPI Code [73] and disclosed in Disclosure UK [74], along with any other institutional recipients of funding that we may not be aware of. Given the importance of small payments in other health contexts [5], the APPG guidelines should be amended so that all payments are disclosed, not just those exceeding £1,500. Under-reporting of industry payments is a consistent problem across various industry payment settings [30,34,57], and APPGs may not be an exception. Payment descriptions also need to be introduced for financial payments and expanded for in-kind payments to give context to the conflicts of interest. To increase transparency around what APPGs actually do, their activities should also be documented in the Register, including links to all published outputs. APPGs, and public bodies more broadly [57], must go further in ensuring that the public to which they are accountable are fully aware of who funds them, why, and the impact [57].

Finally, the payments from patient organisations with conflicts of interest identified in our research suggests that industry might deploy a multi-layered "web of influence" strategy through partnerships with patient organisations. Regulating these indirect types of conflicts is more complicated than direct conflicts of interest as they are inherently hidden [75] as they do

not need to be explicitly reported. Organisations providing payments to APPGs should be required to publicly (on the APPG Register) disclose any corporate funding they have received in the last 12 months, as well as the shares of their income coming from industry [37]. Making this information easily accessible in one place is crucial given the frequent role of industry-funded patient organisations in APPG activities, evidenced by their numerous in-kind payments, and the risk that the patient voice might speak with a 'pharma accent' [76] when involved in policy discussions.

In future research it will also be important to examine industry's ties to other areas of Parliament, such as the relationships between individual policymakers and pharmaceutical companies. Holistically scrutinising industry engagement with influential organisations and individuals is critical to protecting the integrity of policy, strategy, and operational decision-making.

## Supporting information

**S1 Table. Number and value of the pharmaceutical industry's direct financial and in-kind payments.**
(DOCX)

**S2 Table. Number and value of indirect payments from pharmaceutical companies.**
(DOCX)

**S3 Table. Total number and value of direct and indirect payments from pharmaceutical companies at the recipient level.**
(DOCX)

**S4 Table. Total number and value of direct and indirect payments from pharmaceutical companies at the donor level.**
(DOCX)

**S5 Table. Purpose of in-kind payments based on descriptions.**
(DOCX)

**S6 Table. Payments from pharmaceutical industry-funded patient organisations.**
(DOCX)

**S7 Table. Categories of the in-kind payments provided by pharmaceutical industry-funded patient organisations.**
(DOCX)

**S1 Appendix. Preparing the data for analysis.**
(DOCX)

## Acknowledgments

We would like to thank Liz Sheils for carefully reading drafts of the paper. We also extend our gratitude Emma Carmel for her insightful comments and support in developing this paper.

## Author Contributions

**Conceptualization:** Emily Rickard, Piotr Ozieranski.

**Data curation:** Emily Rickard.

**Formal analysis:** Emily Rickard.

**Funding acquisition:** Emily Rickard, Piotr Ozieranski.

**Methodology:** Emily Rickard, Piotr Ozieranski.

**Project administration:** Emily Rickard.

**Supervision:** Piotr Ozieranski.

**Writing – original draft:** Emily Rickard, Piotr Ozieranski.

**Writing – review & editing:** Emily Rickard, Piotr Ozieranski.

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
