## [Decision Letter · Decision Letter 0]

19 Mar 2021

PONE-D-21-06308

A hidden web of policy influence: The pharmaceutical industry’s engagement with UK’s All-Party Parliamentary Groups (2012-2018)

PLOS ONE

Dear Dr. Ozieranski,

Thank you for submitting your manuscript to PLOS ONE. After careful consideration, we feel that it has merit but does not fully meet PLOS ONE’s publication criteria as it currently stands. Therefore, we invite you to submit a revised version of the manuscript that addresses the points raised during the review process.

In addition to the comments from the two reviewers I have some additional points that need to be addressed:

Are there Parliamentary rules or legislation governing the activities of APPGs?Has there been any research into how members of APPGs feel about accepting money from corporate interests?An ethics statement is needed – either that it was obtained or that it was not necessary.Line 106: Explain what is meant by "under the guise of Secretariats".Lines 185-186: Explain further what is meant by "whether industry was involved".

We look forward to receiving your revised manuscript.

Kind regards,

Joel Lexchin, MD

Academic Editor

PLOS ONE

Journal Requirements:

'ER has no conflicts of interests to declare. PO’s PhD student was supported by a grant

from Sigma Pharmaceuticals, a UK pharmacy wholesaler and distributor (not a

pharmaceutical company). The PhD work funded by Sigma Pharmaceuticals is

unrelated to the subject of this paper.'

Additional Editor Comments (if provided):

Reviewers' comments:

Reviewer's Responses to Questions

**Comments to the Author**

1. Is the manuscript technically sound, and do the data support the conclusions?

Reviewer #1: Yes

Reviewer #2: Yes

2. Has the statistical analysis been performed appropriately and rigorously? 

Reviewer #1: No

Reviewer #2: N/A

3. Have the authors made all data underlying the findings in their manuscript fully available?

Reviewer #1: Yes

Reviewer #2: Yes

4. Is the manuscript presented in an intelligible fashion and written in standard English?

Reviewer #1: Yes

Reviewer #2: Yes

5. Review Comments to the Author

Reviewer #1: The authors conducted a cross-sectional study of the payments made from pharmaceutical companies to All-Party Parliamentary Groups in the United Kingdom. This is a novel analysis of a site of industry influence over policy-making and one that is underexplored in relation to health policy making. The authors make use of two publicly available data sets, the UK Parliament’s Register of All-Party Parliamentary Groups and drug company payment disclosure reports, to quantify payments made that pose a conflict of interest for these Groups. They also measure the concept of “confluence of interest” whereby pharmaceutical industry-funded patient groups sponsors the Parliamentary Groups, creating a network of sponsorship.

Major comments:

The authors have found an interesting and important case study to examine industry influence over policymaking. However, I had never heard of All-Party Parliamentary Groups and still do not have a fully clear idea of how they function, nor how these data might translate to other countries. In regard to the transferability of the findings, it would be useful to first position this work within the context of a literature on industry lobbying and its impacts.

Thus, I would suggest reorganizing the introduction so the reader is first introduced to the problem (lobbying/sponsorship, conflict of interest and its impacts on policymaking), and then to the particular case (explaining the political function of All-Party Parliamentary Groups). It would also be helpful to suggest whether there are analogous political bodies in other settings (if they exist). I would also clearly define All-Party Parliamentary Groups and their role in the abstract. My preference as a reader is to avoid acronyms at all costs. Much of this is currently in the Discussion, but should in part, be discussed up front.

The authors sought to provide evidence of “potential conflicts of interest” by identifying “all publications available on their websites and coded them by their type (e.g. report) and indicated whether industry was involved or whether the topics covered might be of interest to industry.” I first suggest eliminating mention of “potential” conflicts of interest throughout; a conflict of interest creates a risk for a primary obligation but it either exists or it doesn’t. However, I think the authors need to first clearly define what a conflict of interest is in the context of All-Party Parliamentary Groups (ie are these pertinent to individuals? Or the institution? What is the primary interest or obligation placed at risk?) to justify whether this actually measures the concept. For example, in section 3.5 why do these publications represent a conflict of interest? It could be that actually what is measured is industry influence, which I think is conceptually distinct from a conflict of interest and conflict of interest should not be used euphemistically to stand in for industry interference in policymaking. You get to this by the Discussion, but I think this should be right up front!

The authors later make recommendations about transparency. However, they also had two public databases at their disposal. Perhaps there could be an added paragraph in the Intro or Methods that provides some context for these data – when/why was the Register created and what data does it include (include S3 Appendix)? Similarly, when/why were the pharma payments made public and what data does this database include?

It would be helpful at the beginning of the Methods to have a small sub-section outlining the overall study design and population from which you sampled.

The Results are difficult to follow. The text is quite dense and the key points are often lost. I would suggest ensuring that no information in the Tables is duplicated in the text. Then, to clearly highlight the key finding for each section. By section 3.3 it is becoming rather repetitive and it is unclear what the added value of parsing the data in these different ways adds. We need more of the author’s analysis to interpret these data, explaining to the reader why it is important, for example, to understand the total impact of relationships #1 and 2 versus these separately, versus payments overall.

There is a lot of descriptive information here, but I think the paper could be strengthened by some comparative analysis. For example, you separately present “direct payments” from the recipient and donor perspective, but the interesting part would be whether this is concordant or discordant. Could you instead triangulate these two data sets and present this information together? What can we learn about direct payments, for example, taking these two data sources together? The numerous supplementary tables that present these data separately (I think?) do not add anything to the analysis; thus, they should be framed as data availability/shared data, or should be further analysed to triangulate the data sources.

Figure 2 provides a nice typology of payment types and can guide the reader through the presentation of the results. However, I would save some of the theorising for the discussion (mechanisms of influence) and just clearly show the different kinds of payments (leaving aside the ‘potential COI’ etc). Then, the results can present: direct financial payments from pharma to Group, indirect payments through third-party, and payments from pharma-funded patient group. My main question regarding this typology is whether the 3rd category (payments from pharma-funded patient groups) is actually a sub-category of the indirect payments? If not, what is the distinction?

The numerous supplementary files include a great deal of material which is essential to understand the case study (e.g. S1 Appendix – this should just be in the introduction; S3 Appendix is necessary to understand the data set – this should be in the intro or methods per my comment above) and also to evaluate the rigour of the methods (the two figures). Having this many supplementary files also creates a great deal of reviewer/reader burden. I suggest that the authors carefully consider what is indeed supplementary and incorporate more of this material into the manuscript (and consider what is essential), or eliminate it. For example S2 Appendix – these terms should be defined in the text, or the authors should use a limited number of these terms to avoid confusion and reduce the need for a glossary. The Supplementary Tables including data seem more appropriate as true supplementary files.

Table S1 (categories of donors) – it seems that the iterative categorisation is not quite finished. This perhaps could be more meaningfully combined into fewer categories. I think this should be guided by a research question and/or framework – i.e. what categories of sponsor are concerning?

Minor comments:

The Methods would be much clearer on first read with the aid of Supplementary Figures 1 and 2 – could these not just be part of the main text (ie not supplementary)? Suggest editing SF2 to eliminate “potential” from conflict of interest.

In the first line of the abstract and in the introduction you mention both “conflicts” and “coincidences” of interests. I am intrigued by this phrasing, but because it is so conceptually interesting, I would appreciate a sentence or two of background, which include definitions of how you conceptualise and differentiate these terms. This is important in both the abstract and main text.

The abstract findings need more contextualising – 16 All-Party Parliamentary Groups received pharma funding, but out of how many in total? Of the sponsoring patient groups, out of how many?

What is the reason for beginning data collection in 2012?

The proportions throughout the text could be rounded to whole numbers or just one decimal place for readability.

Table 1 missing a (%) in the column header “Number of Payments with value – n”

Table 1; you state in the text that Cancer received the highest number of payments, but this is specific to payments from pharmaceutical industry sources. Could you flip the columns and present these on the left hand and the “payments from all external sources” on the right – it would be nice to have an additional column that gives the proportion of pharma payments out of the total received.

The text frequently cross-references itself forcing the reader to jump around, e.g “these APPGs fell into three categories (introduced in Fig 1, reason for inclusion)”; it would enhance readability to just clearly present the necessary information at each point.

Table 2- what is the significance of the bolded text? Are these actually headers? Could this be formatted more clearly?

Thank you for the opportunity to review this manuscript.

Reviewer #2: Major comments:

I definitely would like to see this paper published as it clearly illustrates the potential for the pharmaceutical industry to influence health policy through payments to groups that communicate directly with government, as well as patient organizations. This is a very important topic. However, I think this main message is obscured by all the confusing terminology in the paper, such as “potential” conflicts of interest, “conflicts of coincidence,” “indirect” and “direct” payments, “in-kind” payments, and “relationships” 1, 2, and 3. This paper clearly seems to be talking about financial payments and conflicts of interest. I would not make it more complicated than it needs to be. Also, not all of these terms are used as they commonly are in the COI literature.

My major suggestion is to eliminate all of these terms and just present the data with clear labels as in Table 1 (ie, payments from all external sources, payments from pharmaceutical companies, etc.) and indicate that these payments are a conflict of interest. I provide more specific comments on these terms below.

Abstract:

Need to define, in lay terms, what All-Party Parliamentary Groups are. Even in the abstract, if possible. Can they be summarized as lobbying groups? Try to describe the objectives of these groups in the abstract because context is needed to determine if there is a conflict of interest. In the introduction, the key description is in the first para: “they facilitate engagement between parliamentarians and external organisations, providing expertise on complex policy matters.” This does sound to me what would commonly be called a lobbying group, and this is enforced by the later comments in the Introduction regarding lobbyists acting under the disguise of Secretariats.

“Fifty APPGs also had coincidences of interest, indicated by reporting 304

payments worth £986,054.94 from 57 patient organisations which had received

£27,883,556.3 from pharmaceutical companies across the same period.” It is not clear what coincidence of interest means in this sentence. It’s a COI, patient groups supported by pharma who then support the APPGs. You could say it is an indirect payment that results in the COI.

Introduction:

“Coincidence of interest” is defined in the introduction (line 126) as: “occurring when “a player crafts an array of overlapping roles across organisations to serve his own agenda-or that of his

network-above that of the organisations which he works” [24]. In our study, the ‘player’ is

the pharmaceutical company and the ‘overlapping roles’ are the financial ties to patient

organisations involved in the health policy landscape via APPGs.”

I would not classify financial ties as “overlapping roles.” They are simply payments which are a conflict of interest. I would think of “overlapping roles” as something more insidious, such as pharma companies and patient organizations having the same board members, or a former pharma employee founding a patient organization. It is my understanding that these types of nonfinancial relationships are beyond the scope of this paper.

Financial ties between pharmaceutical companies and patient groups have been extensively investigated. Based on this literature, financial payments to patient groups would clearly fit the definition of a conflict of interest.

Methods

Line 185: “we identified all publications available on their websites and coded them by their type (e.g. report) and indicated whether industry was involved or whether the topics covered might be of interest to industry.” The APPGs cover a wide variety of topics (Fig 1), so this is a very important classification. More information is needed on how the authors determined how “industry was involved” or “how the topics covered might be of interest to industry”

Please explain the overlap in dates of data collection for the topics of the APPG (Which seem to be in 2020, according to lines 180-181) and the payments from pharmaceutical companies, which seem to be much earlier (2012-2018, according to lines 188-193). Should these be more closely aligned?

I did not see the methods describing how patient organization and pharmaceutical company payments to APPGs were determined. Is this in the Register described in lines 151-162? Since payments to the APPGs are a main variable in this paper, the methods for obtaining this information should be clearly described in the body of the text. Lines 207-216 describe the types of payments recorded in the Register and this information should come earlier.

Line 197: It is not clear which database was being screened to eliminate duplicates – the database of payments to APPGs or the database of payments made by pharmaceutical companies?

The classification of direct and indirect payments (lines 222-224) is not what is commonly used in the conflict of interest literature. “Direct” payments usually refer to payments made directly to the person or organization of interest (eg, pharma pays the APPG). “Indirect” payments usually refer to payments that go through a 3rd party (eg, pharma pays a professional society that then pays an APPG). The definition used here: “direct payments involve one single

donor, whereas indirect payments involve at least one other organization” needs clarification. It seems that what is really being calculated is the proportion of payments from a pharmaceutical company when there are payments from multiple organizations. Furthermore, this definition of “indirect” payments does not seem to correspond with the one in S3 which provides details of information in the APPG registries. In S3 states: “APPGs must also name the donor (and any third-party behind the funds if it is indirect)”

RESULTS

Line 244. Provide the denominator for the APPGs. It is not clear how many were actually included in the study. Also, it appears that 2 denominators are used in this paragraph – all health related APPGs and all APPGs reporting payments.

Lines 247 and 249. Again, I find the use of “direct” and “indirect” confusing. Doesn’t the figure for indirect payments mean the proportion of payments from pharmaceutical companies if the payments were from multiple sources? The important piece of information to know is how many and what is the value of payments from pharmaceutical companies to APPGs. I think this is what is meant by “Relationships #1 and #2” in line 263.

Lines 255 – 259. Again, “indirect” and “coincidences” of interest are confusing. How do organizations being funded by pharmaceutical companies to provide a service for an APPG differ from consumer organizations that are funded by pharmaceutical companies and provide payments to APPGs?

Line 259: refers to “pharmaceutical industry funded organizations.” Do we know if the groups are completely funded by pharma or partially funded? This was not clear from the methods.

Table 1 is informative because it clearly states in the column headings where the payments are coming from. However, need to define how the percents are calculated. I believe these are row percents – which clearly shows that the Breast Cancer and Pancreatic Cancer APPGs are 100% funded by the pharmaceutical industry or pharma-supported patient groups, correct?

Please always provide denominators whenever percents are reported in the Results section.

Line 314-315. Mentions “in-kind payments.” These have not been previously defined, please clarify what this means in the Methods section. In-kind payments are mentioned in S2 and S3, but this is not sufficient as this does not explain if they are reported separately or combined with the other payments.

Line 350-51 also mentions “in-kind payments” and gives the example of providing secretariat or administrative services. If no £ values were reported, how were the in-kind services counted? If information on the amount of payment is available, it would be clearer just to report the £ amounts and how the payments were used. My confusion regarding in-kind payments continues throughout the Results section.

Only the APPGs are named in the results section. Why not name the pharmaceutical companies and patient organizations, at least those who are providing the most payments? The pharmaceutical company names are in Table S2 and S3, but it would be useful to at least have the top 10 in the text, especially in relation to the comment below. I did not see the patient organizations named in any supplemental tables.

The section on APPG outputs is important and interesting. But, as per previous comment, it would be useful to know that when the activities which involved drug companies, that these same companies were funding the APPGs (or not). This obviously would not establish any causality, but it would show the financial link between funders of the APPGs and the activities of the APPGs.

Discussion

Line 562 states: “the concentration of payments within a narrow group of donors.” I did not see the list of donors in the Results section.

Minor points:

Data access: Clarify if the Bath Archive makes the data publicly available. Is this open access?

Reference #28 contains extraneous information.

6. PLOS authors have the option to publish the peer review history of their article (what does this mean?). If published, this will include your full peer review and any attached files.

Reviewer #1: **Yes: **Quinn Grundy, University of Toronto

Reviewer #2: **Yes: **Lisa Bero

---

## [Author Response · Author response to Decision Letter 0]

5 May 2021

Please see the file called "responses to reviewers", which forms part of this resubmisison.

---

## [Decision Letter · Decision Letter 1]

18 May 2021

A hidden web of policy influence: The pharmaceutical industry’s engagement with UK’s All-Party Parliamentary Groups

PONE-D-21-06308R1

Dear Dr. Ozieranski,

We’re pleased to inform you that your manuscript has been judged scientifically suitable for publication and will be formally accepted for publication once it meets all outstanding technical requirements.

Kind regards,

Joel Lexchin, MD

Academic Editor

PLOS ONE

Additional Editor Comments (optional):

Reviewers' comments:

Reviewer's Responses to Questions

**Comments to the Author**

1. If the authors have adequately addressed your comments raised in a previous round of review and you feel that this manuscript is now acceptable for publication, you may indicate that here to bypass the “Comments to the Author” section, enter your conflict of interest statement in the “Confidential to Editor” section, and submit your "Accept" recommendation.

Reviewer #1: (No Response)

2. Is the manuscript technically sound, and do the data support the conclusions?

Reviewer #1: Yes

3. Has the statistical analysis been performed appropriately and rigorously? 

Reviewer #1: Yes

4. Have the authors made all data underlying the findings in their manuscript fully available?

Reviewer #1: Yes

5. Is the manuscript presented in an intelligible fashion and written in standard English?

Reviewer #1: Yes

6. Review Comments to the Author

Reviewer #1: The authors have thoroughly addressed all my previous comments. In particular, the introduction is much strengthened and provides a compelling rationale for the study and a clear description of the case under study and the results are much more impactful. Thank you for the opportunity to review this manuscript and I look forward to seeing it published!

7. PLOS authors have the option to publish the peer review history of their article (what does this mean?). If published, this will include your full peer review and any attached files.

Reviewer #1: **Yes: **Quinn Grundy, University of Toronto

---

## [Editor Report · Acceptance letter]

14 Jun 2021

PONE-D-21-06308R1 

A hidden web of policy influence: The pharmaceutical industry’s engagement with UK’s All-Party Parliamentary Groups 

Dear Dr. Ozieranski:

I'm pleased to inform you that your manuscript has been deemed suitable for publication in PLOS ONE. Congratulations! Your manuscript is now with our production department. 

Kind regards, 

on behalf of

Prof. Joel Lexchin 

Academic Editor

PLOS ONE